# Injectable Nano Drug Delivery Systems for the Treatment of Breast Cancer

**DOI:** 10.3390/pharmaceutics14122783

**Published:** 2022-12-13

**Authors:** Urmila Kafle, Satish Agrawal, Alekha K. Dash

**Affiliations:** Department of Pharmacy Sciences, School of Pharmacy and Health Professions, Creighton University, 2500 California Plaza, Omaha, NE 68178, USA

**Keywords:** breast cancer, metastasis, nanomedicine, injectable, controlled release, organic nanoparticles, inorganic nanoparticles, nano-drug delivery systems

## Abstract

Breast cancer is the most diagnosed type of cancer, with 2.26 million cases and 685,000 deaths recorded in 2020. If left untreated, this deadly disease can metastasize to distant organs, which is the reason behind its incurability and related deaths. Currently, conventional therapies are used to treat breast cancer, but they have numerous shortcomings such as low bioavailability, short circulation time, and off-target toxicity. To address these challenges, nanomedicines are preferred and are being extensively investigated for breast cancer treatment. Nanomedicines are novel drug delivery systems that can improve drug stability, aqueous solubility, blood circulation time, controlled release, and targeted delivery at the tumoral site and enhance therapeutic safety and effectiveness. Nanoparticles (NPs) can be administered through different routes. Although the injectable route is less preferred than the oral route for drug administration, it has its advantages: it helps tailor drugs with targeted moiety, boosts payload, avoids first-pass metabolism, and improves the pharmacokinetic parameters of the active pharmaceutical ingredients. Targeted delivery of nanomedicine, closer to organelles such as the mitochondria and nuclei in breast cancer, reduces the dosage requirements and the toxic effects of chemotherapeutics. This review aims to provide the current status of the recent advances in various injectable nanomedicines for targeted treatment of breast cancer.

## 1. Introduction

Cancer arises when normal cells convert into tumor cells in a multi-stage process that generally begins with a pre-cancerous lesion. This transformation is caused by the interplay of a person’s genetic outcome and environmental variables (physical, chemical, and biological carcinogens). According to the WHO, cancer is one of the major global causes of mortality, leading to the death of nearly 10 million people in 2020 [1]. Breast cancer is the leading cancer, with 2.26 million cases recorded in 2020, and the most common cause of female cancer deaths, with 685,000 recorded [2,3].

Breast cancer is a deadly disease that can metastasize to distant organs such as the bones, liver, lungs, and brain. This type of metastasis is usually the cause of breast cancer’s incurability. If detected early, breast cancer has an excellent prognosis and a high survival rate. Breast cancer is exacerbated by risk factors such as sex, age, family history, and an unhealthy lifestyle [4]. According to the National Institutes of Health, in 2022, the estimated number of new breast cancer cases in the USA is 287,850, accounting for approximately 15% of all new cancer cases. Similarly, the death rate from breast cancer is 7.1% of all the cancer deaths recorded [5]. In the next 20 years, incidence and mortality rates are anticipated to rise dramatically by 55% and 58%, respectively, in developing nations [6,7].

The treatment options for breast cancer depend on its type and the stage at which it is first diagnosed. Currently available options include chemotherapy, surgery, immunotherapy, gene therapy, and hormonal therapy. Among these, chemotherapy is the most preferred method for cancer treatment. Chemotherapy is defined as the use of cytotoxic agents to prevent cancer cells from growing and spreading and eliminate them. However, chemotherapeutic agents are equally responsible for unwanted toxicity to normal tissues. The majority of chemotherapeutics come with the disadvantages of low bioavailability, the short circulation time of their active pharmaceutical ingredients, and toxicity to normal cells. Injectable nanomedicines are being explored to overcome these challenges, to increase effectiveness while reducing potential side effects [8].

Different modes of administration, such as oral, injectables (intravenous (IV), intraperitoneal, and intramuscular), and pulmonary inhalation, are all viable options for the delivery of nanomedicines. Although the injectable route is less preferred than the oral route, it still offers certain benefits that can be beneficial for patients with critical condition such as cancer. The injectable route can provide an instant response, increased bioavailability, an active targeting ability, and increased efficacy and avoid first-pass metabolism [9]. 

Nanomedicines are novel approaches to drug delivery systems, designed by controlling their size and shape in the nanoscale range. The choice of nanomedicine size for a cancer therapy involves finding a balance between maximizing tumor penetration (by reducing its size) and minimizing both toxicity to normal tissues and clearance by the mononuclear phagocyte system (by increasing its size) [10]. Depending on the desired properties, it can also be smaller than 100 nm or up to 800 nm. Nanomedicines may be scaled to multiple sizes, shapes, and chemical compositions, and they offer a high surface-area-to-volume ratio and adjustable optical characteristics [11,12]. 

Nano-delivery systems also help to improve drug stability, aqueous solubility, blood circulation time, controlled release, and the targeted delivery of the drugs at the tumor site. Additionally, nanomedicines assist in drug accumulation at the tumor either by active or passive mechanisms. Passive mechanisms utilize an enhanced permeation and retention (EPR) effect that exploits the leaky nature of the tumor vasculature and the prolonged circulation of nanomedicines, allowing a slow but preferential accumulation in the tumor bed [13]. The active targeting mechanisms aim to target the site of action using targeting moieties on the nanoparticle (NP)’s surface [14,15].

## 2. Breast Cancer Biology

Breast cancer is a fatal illness with an appearance that differs among women [16]. There are three types of breast cancer based on molecular and histological findings: breast cancer expressing hormone receptors (estrogen receptor positive (ER^+^) or progesterone receptor positive (PR^+^)), breast cancer expressing human epidermal growth factor receptor 2 (HER2^+^), and triple-negative breast cancer (TNBC) (ER^-^, PR^-^, HER2^-^) [17,18]. Usually, the treatment approach for breast cancer depends on the tumor’s molecular characteristics. In addition, TNBC is subdivided into six subtypes, as shown in Table 1, which include immunomodulatory, mesenchymal, luminal androgen receptor, mesenchymal stem cell-like, basal-like 1, and basal-like 2 [18].

The exact cause of breast cancer remains unclear. Hormone-receptor-expressed breast cancer occurs the most among all three forms of breast cancer, with an incidence of 60–70%, especially in premenopausal women. Hormone therapy is used for its treatment, which works by preventing hormone production from the ovary [22]. The commonly used drugs are estrogen blockers (tamoxifen) and aromatase inhibitors (anastrozole, exemestane) [23].

Chemotherapies and anti-HER2 monoclonal antibodies (trastuzumab, pertuzumab) are the mainstay therapy for HER2^+^ breast cancer [24]. TNBC is the challenging breast cancer among others for treatment. The standard therapy for TNBC is chemotherapy. However, there are other ways to treat TNBC, such as using chemotherapy along with a recombinant humanized monoclonal antibody against vascular endothelial growth factor [17,25].

### 2.1. Risk Factors

**a.** **Age:** With growing age, the risk of developing breast cancer increases. According to the surveillance, epidemiology, and end results databases, a woman in the United States has a 1 in 8 chance of being diagnosed with breast cancer in her lifetime: 1 in 202 from birth to age 39, 1 in 26 from age 40 to 59, and 1 in 28 from age 60 to 69 [26].**b.** **Breast Pathology:** Proliferative breast disease increases the probability of developing breast cancer. Proliferative breast lesions devoid of atypia are just marginally more likely to progress to breast cancer [27]. When atypical hyperplasia is discovered by mammography screening, it suggests a considerable risk for the occurrence of breast cancer. Women with atypia are 4.3 times more likely to be diagnosed with breast cancer than the general population [27,28].**c.** **Family History:** There is a correlation between breast cancer in a woman’s family and her own risk of developing the disease. The possibility is highest if the first-degree relatives of the women have been recognized with breast cancer at an early age. Family history is useful for identifying individuals who may be carriers of a genetic mutation predisposing to breast cancer, such as breast cancer gene-1 or gene-2 [29].**d.** **Early Menarche:** Menarche is the first occurrence of menstruation. Early-age menarche increases the chances of developing breast cancer in both pre- and post-menopausal women compared to women with delayed menarche [30].**e.** **Bone Density:** Bone contains estrogen receptors, so increasing bone density may function as an alternative indicator for circulating estrogen and is linked with a higher risk of breast cancer [31]. According to a meta-analysis, women with a high hip bone density were more likely to be diagnosed with breast cancer [32].**f.** **Breast Density:** Breast density measures the proportion of fibro glandular tissue. If the breasts have a fibro glandular tissue content higher than 75%, then they are considered mammographically dense. Women with dense breasts are more likely to be diagnosed with breast cancer. The chances are even higher if a woman is under the age of 56 [33,34,35].**g.** **Lifestyle:** An increase in weight during the perimenopausal period is associated with an increased risk of breast cancer [36,37,38]. Physical activity demonstrates a protective impact by lowering hormone levels [34]. Similarly, increased alcohol use and smoking habits are linked with an increased breast cancer risk [39,40,41].**h.** **Breastfeeding:** Women who breastfeed have a lower possibility of developing breast cancer. Lactation slows normal ovulatory cycles and reduces endogenous sex hormone levels.**i.** **Age of Menopause:** Delayed menopause is related with an increased risk of being diagnosed with breast cancer.

### 2.2. Stages of Breast Cancer

**a.** 
**Stage 0: In Situ**


A secondary microscopic discovery of abnormal tissue development in the lobular breast is called in situ lobular cancer. Neither breast progresses to another invasive breast cancer, but the probability of a successive invasive breast cancer increases by about 7% over 10 years [42]. In in situ lobular carcinoma, local and systemic therapies are not given. Instead, these women need to be monitored closely for signs of breast cancer. Patients are taught about selective estrogen receptor modulators (SERMs) such as tamoxifen for use as a chemical prophylaxis.

**b.** 
**Stage I and Stage II: Early-Stage Invasive**


Stage I is the earliest stage of invasive breast cancer, where the tumor size ranges up to 2 cm without affecting any lymph nodes. The cancer cells spread to a distant location from the original location, which surrounds the breast tissues [43].

In stage II, also known as invasive breast cancer, the tumor may be 2–5 cm in size. It usually spreads to the lymph nodes under the arm. Stage II breast cancer presents with a slightly new type of the disease. At this stage, cancer cells migrate outside the breast tissues, and the tumor becomes bigger than in stage I. However, stage II indicates that the disease has not spread to distant organs [43].

**c.** 
**Stage III: Locally Advanced**


Stage III is locally advanced breast cancer, and the tumor size of the breast tissue exceeds 5 cm in diameter. It is extensively found in the underarm lymph nodes. It is a new form of invasive breast cancer, and cancer cells usually do not spread further in the body. The tumor is huge at this stage, which may extend to the chest wall or the skin of the breast [43].

**d.** 
**Stage IV: Metastatic**


Stage IV occurs when the tumor has spread to distant parts of the body. The affected areas include the brain, liver, bones, and lungs. The cancer can spread to more than one part of the body [43].

**TNM staging** helps to provide detailed information about the stages of breast cancer [43].

**(A).** **T:** It refers to the size of the tumor. Tumors can be measured using imaging techniques.
**Tx**: No evaluation of a tumor.**T0**: No evidence of a tumor.**Tis**: Cancerous cells were detected prior to tumor development.**T1**: Tumor is less than 2 cm.**T2**: Tumor is between 2 and 5 cm.**T3**: Tumor is bigger than 5 cm.**T4**: Tumor with any size that grows into the chest wall or the skin.**(B).** **N:** It is the proliferation of the tumor to the nearby lymph nodes. Cancer found in the lymph node can be minute, called micrometastasis (0.2 mm to 2 mm), or big, called macrometastasis (bigger than 2 mm).
**Nx**: No evaluation of lymph nodes.**N0**: No indication of any spread.**N1**: Cancer has spread in smaller amounts to nodes near the breastbone or underarm lymph nodes.**N2**: Cancer has spread in larger quantities to the underarm lymph nodes than in N1.**N3**: Cancer has spread significantly to the underarm lymph nodes.**(C).** **M:** It is the spread of the tumor to several distant parts of the body.
**Mx**: No evaluation of metastasis.**M0**: No evidence of any spread.**M1**: Spread of cancer to distant organs or tissues.



**Additional Markers for Breast Cancer Staging**


For the targeted treatment of breast cancer, additional markers specific to breast cancer subtypes can be helpful [43].

**ER**: Breast cancers have a receptor that responds to the estrogen hormone.**PR**: Breast cancers have a receptor that responds to the progesterone hormone.**HER2**: Breast cancer makes an excess amount of the protein HER2.**G**: It is the grade of cancer to differentiate cancerous cells from normal cells.
⮚**Grade 1:** Cells appear uniform.⮚**Grade 2:** The rate of cell division increases.⮚**Grade 3:** Cells vary in appearance from normal breast tissue.

## 3. Treatment Options

Patients with breast cancer in stages III and IV have a bad prognosis. The survival rates for such patients are only 3–5 years. Similarly, 10% of newly diagnosed breast cancers progress to metastatic disease. Additionally, patients diagnosed at earlier stages have a 50% chance of developing metastatic breast cancer [44]. To prevent such an occurrence, several conventional treatment approaches are used.

### 3.1. Conventional Approaches for Breast Cancer Treatment

**a.** 
**Surgical Management**


The conventional treatment for early-stage breast cancer often involves breast amputation and then exposing the patient to radiation [45]. Though conservative treatment helps to achieve a comparable survival rate, it also offers higher chances of local recurrence [46]. A prophylactic bilateral mastectomy is also a method of surgical management for breast cancer. Women use this method at their own risk, as there is no clear evidence that it will help young people with breast cancer who do not have the breast cancer gene-1 mutation [44].

**b.** 
**Adjuvant Radiotherapy**


Radiotherapy is usually given after breast-conserving surgery and sometimes after a mastectomy, especially when the tumors are larger (>5 cm). Radiotherapy is also given when breast cancer metastases have occurred to at least four lymph nodes. Younger women are at a greater risk of breast cancer recurrence, even after breast-conserving surgery or a mastectomy; therefore, radiation gives them more recurrence-prevention advantages than older women [47,48].

**c.** 
**Hormonal Therapy**


Hormonal therapy is extensively used in the adjuvant treatment of early stages of breast cancer and specialized care for metastasized disease in young women. This therapy mainly works in women whose breast cancer shows steroid receptor expression. However, hormonal therapy is not indicated for neoadjuvant therapy due to a lack of efficacy data [44].

Tamoxifen is the most used hormonal therapy for breast cancer treatment. The International ATLAS (Adjuvant Tamoxifen Longer Against Shorter) breast cancer therapy study indicated that tamoxifen treatment for 10 years after the original diagnosis reduces the risk of death by one-third, and beyond 10 years it reduces mortality by half [49]. On the contrary, one study observed that the risk of developing cervical cancer increased by twofold in patients who were given tamoxifen for 10 years. In hormone-dependent cancers, there will still be a 2–3% risk of recurrence because the survival curve never reaches a plateau with these therapies [44].

**d.** 
**Chemotherapy**


Chemotherapy is an anti-cancer drug treatment that can be given by oral or parenteral routes [50]. The drugs administered via the IV route are directly released in the bloodstream to reach their target. Depending on the situation, various kinds of chemotherapy are recommended.

**i.** 
**Adjuvant Chemotherapy (After Surgery)**


The primary objective of adjuvant chemotherapy is to eliminate cancer cells that may have been neglected or were not detected by imaging testing. It helps to reduce the risk of breast cancer recurrence. Some tests, such as Oncotype DX, which predicts how likely breast cancer is to spread to somewhere else in the body, are used to determine whether adjuvant chemotherapy will be helpful or not [50]. Adjuvant chemotherapy is recommended for the following conditions:Cancer with a triple-negative phenotype;Cancer with ER expression (chemotherapy given with hormonal therapy);Cancer with HER2 overexpression (anthracycline chemotherapy preferred with monoclonal antibodies).

**ii.** 
**Neoadjuvant Chemotherapy (Before Surgery)**


Neoadjuvant chemotherapy is used to shrink the size of the tumor, so it may be removed with minimally invasive surgery. When tumors are too big, have many lymph nodes, and are inflammatory, neoadjuvant chemotherapy is usually given [50]. Neoadjuvant chemotherapy can be given because of the following benefits it provides:Extra time for genetic testing;Increased longevity of a patient with early-stage cancer, especially with TNBC or HER2-positive breast cancer.

**iii.** 
**For Metastatic Breast Cancer**


In metastatic cancer, cancer cells proliferate from the breast area to distant organs such as the lungs or the liver. Chemotherapy is the major treatment for metastatic breast cancer. It can be given directly after the diagnosis of breast cancer or after treating the cancer cells with neoadjuvant therapies. The length of chemotherapy treatment is usually patient-specific [50]. Some of the chemotherapeutic agents and their doses used in breast cancer treatment are shown in Table 2.

### 3.2. Challenges of Using Conventional Therapy and Their Remedies

The pharmacological agents that are used in conventional treatment methods are not selective and can cause off-target delivery of the drugs. Due to off-target delivery, a high concentration of the drugs will be needed to achieve the same level of therapeutic effects, which leads to problems such as toxicity to cancer cells and multi-drug resistance [59,60]. Similarly, cancer is very heterogeneous, containing several cancerous cells and stem cells. While standard treatments can reduce the size of a tumor, they do little to eliminate the cancerous stem cells that fuel relapses and spread.

The problems associated with multi-drug resistance can be reduced with the use of multi-drugs, which act on the breast cancer receptors by different mechanisms. The use of multi-drugs also helps to maintain the dosage of the drugs within the therapeutic window [61,62]. Even though combination drugs help solve the problem of drug resistance, they still do not target cancer cells and have low bioavailability. To overcome these challenges, nano-drug carriers can be applied to combinational deliveries. Using nanocarriers increases the bioavailability of drugs at the target site, decreases drug degradation by circumventing the reticuloendothelial system, and even decreases off-target toxicity. In addition to this, nano-drug delivery systems can help in the co-delivery of multi-drugs in a controlled manner, which will help to maintain the normal pharmacokinetics and pharmacodynamics characteristics of multiple therapeutic agents [63,64].

## 4. Injectable Nano-Drug Delivery Systems

The development of injectable drug delivery systems has received considerable attention over the past few years. For the injectable route of administration, the drugs are administered at different sites, e.g., IV, intramuscular, subcutaneous, etc. Although this route of administration has some disadvantages such as pain at the site of injection, risk of embolism, strict adherence to aseptic procedures, and difficulty in self-administration, the advantages outweigh their disadvantages for use in life-threatening diseases such as cancers. The drugs directly reach the blood, bypassing the gastrointestinal tract and enabling very rapid and high (100% in case of IV) absorption of the drugs. Moreover, this route is precise and accurate and has an almost immediate onset of action, so a high dose of drugs can be administered, fairly pain free. This route also makes it possible to administer medications to unconscious patients and patients with vomiting and diarrhea or drugs with an unpleasant taste [65,66,67].

Currently, there is an increased use of nanotechnology to deliver anticancer agents via this route. Injectable nanosized-drug delivery systems possess several advantages over other routes of administration, which include an improvement in the solubility profile of poorly soluble drugs and a reduction in the metabolism of anticancer agents by dissolving them in the hydrophobic and hydrophilic core of the NPs. Furthermore, nano-drug delivery systems show EPR effects, which help in the passive targeting by anticancer agents of the tumor. In addition, nanomedicines have a high surface-to-volume ratio for drug loading, a size that can be modified with a longer plasma half-life, and a more distinct biodistribution profile than traditional chemotherapy [68]. Presently, there are several anticancer-agent-based nano-formulations approved by the FDA for clinical application, and they are administered via intravenous route and are listed in Table 3. The nano-formulations undergoing clinical trials and at different phases are presented in Table 4, providing hope and a potential bright future for nanomedicines in cancer treatment [69]; all these formulations are being administered via injectable routes.

## 5. Types of Nano-Drug Delivery Systems

The delivery of anticancer drugs can be accomplished via a variety of nanotechnology techniques. All these methods have the capacity to modify active targeting ligands and have distinctive physiochemical features of their own. This review discusses how different types of NPs can encapsulate therapeutic ingredients and achieve targeted delivery via the injectable route to treat breast cancer. The results from in vitro studies are also discussed because of their potential to be administered via the injectable route. 

### 5.1. Organic Nanoparticles (NPs)

Organic NPs are amphiphilic lipid molecules, which can self-assemble in an aqueous environment. These drug delivery systems can incorporate both hydrophilic and hydrophobic drugs and can provide sustained release. Figure 1 depicts the different types of organic NPs that can be used in breast cancer treatment, and their advantages over conventional dosage forms are described in this section.

#### 5.1.1. Micelles

Polymeric micelles are colloidal particles composed of amphiphilic copolymers that self-assemble in an aqueous medium. Micelles contain a hydrophobic core and hydrophilic surroundings, which aid in loading multiple drugs (hydrophilic and hydrophobic) simultaneously. Micelles can accumulate in poorly vascularized tumors and potentiate the EPR effect of NPs. In addition, micelles are observed to reduce P-glycoprotein efflux and raise intracellular drug concentration in Doxorubicin (Dox)-resistant, highly cytotoxic MCF-7 cells [76,77].

Poly (ethylene glycol)–polylactide acid is one of the most used, surface-modified biodegradable polymers in the development of drug delivery systems such as micelles and polymeric NPs due to its exceptional physicochemical and biological properties, which include non-toxicity, the exemption of protein adsorption, and resultant non-specific absorption by the reticuloendothelial system after an IV injection [78]. 

Genexol-PM is an FDA-approved polyethylene glycol-polylactide (PEG-PLA) polymeric micelle of paclitaxel used to treat breast cancer [70]. Genexol-PM showed a higher anti-tumor activity and more accumulation in tumor tissue compared to conventional paclitaxel therapy. The phase II study designed to evaluate and study the safety and efficacy of Genexol-PM in patients with metastatic breast cancer administered as an IV infusion observed an overall 58.5% increase in the response rate in comparison to the conventional paclitaxel therapy [79].

Poly (HEMA-LA-MADQUAT) micelles were recently developed for the co-delivery of methotrexate and chrysin administered by IV route for the treatment of breast cancer. The copolymer in this study was synthesized by using two different polymerization process that include ring-opening polymerization and free-radical polymerization. The drug loads of both drugs in the polymeric micelles were more than 85%. The in vitro evaluation of these polymeric micelles on the MCF-7 breast cancer cell line demonstrated a better anticancer efficacy than free anticancer drugs (methotrexate and chrysin) [80].

#### 5.1.2. Liposomes

Liposomes are spherical vesicles with an aqueous interior and lipid bilayers on the exterior. They can load both hydrophilic and hydrophobic drugs while protecting them from the external environment [81]. A liposome’s diameter typically ranges from 50 to 200 nm, and they can accumulate in tumor cells with an improved EPR effect. Liposomes are stable, non-immunogenic, biodegradable, biocompatible, and non-toxic. They are categorized according to their size and composition: small unilamellar vesicles, large unilamellar vesicles, and multilamellar vesicles. The characteristics of these liposomes are affected by several variables, including preparation techniques, surface charge, and surface hydration [82].

To combat resistant breast cancer, liposomal formulations of larotaxel were adorned with a lipid derivative containing guanine-rich quadruplex nucleotides [83]. The nucleotide-lipid derivative [1, 2-Distearoyl-sn-glycero-3-phosphoethanolamine-Poly (ethylene glycol)2000 -C6 -GT28nt] was prepared by including a hydrophobic hexyl linkage between GT-28nt (containing 17 guanines and 11 thymidines) and 1, 2-Distearoyl-sn-glycero-3-phosphoethanolamine-Poly (ethylene glycol)200-N-hydroxysuccinimide and then included on the functional larotaxel liposomes for specific binding with the nucleolin receptor on the resistant MCF-7/ADR cancer cells. These liposomes, administered via the IV route, showed a higher anticancer efficacy, increased circulation, and targeted effects to the resistant breast cancer cells. The formulation showed the advantages of blocking the depolymerization of the microtubules, inhibiting the activity of the antiapoptotic proteins, and blocking the Janus kinase/signal transducers and activators of the transcription pathway. Therefore, larotaxel liposomes can be a unique therapy for breast cancer that is resistant to many drugs.

Similarly, Wang et al. [84] developed a liposomal formulation of acetyltanshinone II A, commonly used to treat ER^+^ breast cancer. PEG-modified liposomes were prepared to entrap acetyltanshinone II A. A single IV injection administered to rats indicated that liposomal ATA has 59 times more bioavailability than free acetyltanshinone II A alone. The preclinical investigations further revealed that it inhibited the development of ER^+^ breast cancers by 73% in nude mice while exhibiting a lesser toxicity to healthy cells. As a result, the bioavailability and therapeutic effectiveness of liposomal acetyltanshinone II A modified with PEG improved the treatment of ER^+^ breast cancer.

#### 5.1.3. Dendrimers

In 1917, Vogtle first introduced dendrimers as nano-drug delivery systems [85]. Dendrimers are highly branched three-dimensional structures possessing a low polydispersity index [82]. The layers formed between each cascade point of the branches in dendrimers are called “generations” [85]. Dendrimers’ surface modification and binding to ligands can control their size and molecular mass [86]. It is possible to load and bind both hydrophilic and hydrophobic drugs using these dendrimeric NPs and inject them.

In 2019, Guo et al. [87] developed hyaluronic acid-modified amine-terminated fourth-generation polyamidoamine (PAMAM) dendrimer NPs for the co-delivery of cisplatin and Dox. The developed dendrimers can enter cells in a time-dependent manner via the lysosome-mediated route, as shown in in vitro studies. According to cell viability experiments, dendrimers showed a potent anticancer effect on MCF-7 and MDA-MB-231 breast cancer cells even at low concentrations of the drugs. Along with effectively reducing tumor growth, dendrimers also reduced Dox’s toxicity. Furthermore, the IV dendrimer treatment of BALB/c nude mice with the MDA-MB-231 tumor resulted in the accumulation of drug-loaded dendrimers at the tumor site, remarkably suppressing tumor growth without any toxicity to normal cells. These findings suggest that conjugated dendrimers can greatly improve the efficacy of cisplatin and Dox as chemotherapeutic agents for breast cancer treatment.

Dendrimers are also widely used in gene delivery. Jain et al. [88] developed dendriplexes by complexing third-generation phosphorous and fourth-generation PAMAM dendrimers with polo-like kinase small interfering RNA (siRNA), also known as siPLK therapies. Thus, the dendriplexes produced in the presence of ribonuclease and serum were stable. They helped increase the cellular uptake of siPLK compared to siPLK solution in MDA-MB-231 and MCF-7 cell lines. Furthermore, dendriplexes of siPLK also increased cell arrest in sub-growth phase1 compared to its solution form. These results imply that phosphorous and PAMAM dendrimers are promising siPLK delivery vehicles for the treatment of TNBC.

#### 5.1.4. Polymeric NPs

Polymeric NPs are sub-micron colloidal systems prepared by binding a copolymer to a polymer matrix. Polymeric NPs utilize dissolution, entrapment, adsorption, and binding techniques for loading drugs [89]. Polymers widely used in NPs formulation are classified as natural, synthetic, bio-degradable, and non-biodegradable. Commonly used natural polymers are cellulose, alginate, and gelatin, which show mild immunogenic behavior [82]. Synthetic polymers such as PLA, and poly lactic-co-glycolide offer higher solubility and permeability. These polymers are biocompatible and biodegradable, with better drug release and stability [90].

Based on the NP’s preparation method, they can be classified as nanocapsules and nanospheres. Nanospheres are matrix systems in which the drug is physically and evenly dispersed. In contrast, nano capsules are vesicular systems in which a drug is contained in a cavity enclosed by a polymer membrane [59]. With proper design, polymeric NPs can help target tissues and tumor cells [91]. Additionally, polymeric NPs utilize their submicron size to accumulate drugs and promote the sustained release of drugs with the aid of biodegradable constructive materials [89].

Using polymeric NPs and light, Jadia et al. [92] developed a remotely triggered targeted treatment to treat TNBC. As TNBC overexpresses the transferrin receptor, the NPs conjugated to a peptide (human transferrin) can actively target TNBC by binding to transferrin receptors. Simultaneously, phototherapy was activated by triggering benzoporphyrin derivative monoacid, a photosensitizer, using near-infrared light. In vitro imaging and cytotoxicity were used to investigate the use of actively targeting polymeric NPs for photodynamic therapy against TNBC. In this study, the fluorescence image confirmed that benzoporphyrin-derivative monoacid-loaded NPs given via the IV route showed a more significant enhancement in the fluorescence of the TNBC cells than its free form. The maximum photo-triggered cytotoxicity was demonstrated by actively targeting NPs in TNBC cells, making them a promising candidate for TNBC treatment.

Similarly, Nabi et al. [93] developed gefitinib mucin-1 aptamer-conjugated NPs using PAMAM to treat breast cancer cells and tumors utilizing image guidance. To create image-guided nanoplatforms, aptamer-conjugated NPs were radiolabeled using gallium-67 as an imaging agent. The sustained-release mechanism of the drug was observed for more than 7 days and supported the kinetic release models of gefitinib from radiolabeled NPs. In vitro testing revealed increased absorption and higher cytotoxicity of the mucin-grafted NPs in MCF-7 cells. These results suggest that the modified injectable NPs are a potential nuclear medicine-assisted image-guided nanosystem for breast cells and tumors that express mucin.

Likewise, Ying et al. [94] developed macrophage membrane biomimetic adhesive polycaprolactone nanocamptothecin to improve targeting efficacy and reduce metastasis for treating breast cancer. In this study, the parent camptothecin drug, called 7-ethyl-10-hydroxy-camptothecin, was revitalized by the polymer conjugation, which resulted in increased cellular uptake and tumor tropic effects. The macrophage camouflaged polycaprolactone nanocamptothecin given via the IV route and showed a more significant amount of tumor accumulation than the uncoated NPs, when it was evaluated in a preclinical murine model of breast cancer. Similarly, a histological examination of the tumor tissues revealed that the destruction of tumor cells was more significant when macrophage-coated NPs were used as opposed to uncoated NPs. A dramatically high degree of intratumor apoptosis was seen because of treatment with polycaprolactone nanocamptothecin NPs. This study indicates that polymeric prodrug design combined with a cell membrane cloaking process can help to achieve a high efficacy and low toxicity against breast cancer treatment.

Ormeloxifene-loaded PEGylated chitosan NPs were developed and optimized for enhancing therapeutic activity against breast cancer. In this study, NPs were prepared by the ionotropic gelation method. The NPs had 88.4% entrapment and 21.0% loading efficiency for ormeloxifene. Additionally, compared to free ormeloxifene, injectable NPs displayed a dose-dependent increase in cytotoxicity, cellular uptake, apoptosis, and activation of caspase-3 in the MDA-MB-231 and MCF-7 cell lines. In vivo investigation when administered via the IV route showed that female Sprague Dawley rats treated with NPs had enhanced pharmacokinetic characteristics, decreased toxicity, a lowered tumor burden, and increased survival rates. This work demonstrated the potential of ormeloxifene NPs encapsulated in PEGylated chitosan NPs as a treatment for breast cancer [95].

#### 5.1.5. Drug Nanocrystals

Drug nanocrystals are pure drug NPs that are stabilized by surfactants or polymeric steric stabilizers with increased drug-loading, due to the absence of any carrier system, and an ease of administration by different routes (e.g., oral, parenteral). The high drug load, superior structural stability, steady dissolution, and prolonged circulation time of drugs facilitated by the nano-crystallization technique have increased research on nanocrystals to treat breast cancer in recent years [19,96]. Drug nanocrystals are usually prepared by using three different approaches. The top-down approach includes a particle size reduction in coarse drugs to nanoscale form either by milling or high-pressure homogenization [97]. The second approach is the bottom-up method, which is a low energy process primarily utilized for thermolabile drugs [98]. The precipitation method is a classic example of the bottom-up method for preparing nanocrystals. The third method is a combination approach, which uses both the bottom-up and top-down approaches for preparing nanocrystals.

Zhang et al. [99] studied the effects of PEGylated paclitaxel nanocrystals on breast cancer and its lung metastasis and prepared paclitaxel nanocrystals with and without PEGylation by using antisolvent precipitation technique. The nanocrystals formed were nano in size with a higher stability in both storage and their physiological conditions. When both nanocrystals were evaluated in xenografted mice via the IV route, PEGylated nanocrystals showed a higher tumor inhibition. Similarly, PEGylated nanocrystals showed a higher anticancer efficacy in the lung tumor metastasis model. This study suggests the potential advantages of PEGylated paclitaxel nanocrystals as an alternative drug delivery to treat breast cancer and its lung metastasis.

#### 5.1.6. Exosomes

Exosomes are membrane-bound extracellular vesicles found in body fluids, which are usually nanoscale in size [100]. They help transport cellular byproducts such as proteins, lipids, DNA, and RNA that modulate intercellular communication in physiological and pathological settings [101,102,103]. In breast cancer, exosomes play a vital role in the two-way communication between tumor cells and the tumor microenvironment [104]. Proteome and genome profiling studies have revealed that the composition and number of circulating exosomes vary between breast cancer patients and healthy controls. As a result, they can be used as a potential source of tumor information [105,106].

Tumor cells often release more exosomes than normal to communicate with local or far-off cells in the tumor microenvironment. Exosomes have a significant role in nearly all of cancer’s key characteristics, such as immune system evasion, genomic instability, apoptosis resistance, unregulated cellular energetics, invasion, and metastasis [107]. Exosomes possess diagnostic and therapeutic potential for the treatment of breast cancer. Tumor-derived exosomes from patients can be found in ascites fluid, urine, pleural effusions, and serum. Since the exosomal cargo reflects the tumor state, researchers can evaluate breast-tumor-derived exosomes to create diagnostic methods that will aid in monitoring disease growth, treatment efficacy, and resistance mechanisms [103,105,108,109]. 

Exosomes can also be used to deliver drugs, as they contain membrane proteins that aid recipient cells in endocytosis, prevent phagocytosis by monocytes [110], and deliver anticancer drugs, proteins, and nucleic acids to the target cells without causing systemic toxicity [111]. They are biocompatible, non-cytotoxic, and can hold more cargo for targeted distribution [112,113,114]. So it would be more pragmatic to utilize customized NPs that incorporate the favorable properties of natural exosomes [114].

Zhao et al. [115] studied exosome-mediated siRNA delivery for the suppression of postoperative breast cancer metastasis. This study identified exosomes from breast cancer cells with practical lung-targeting ability. They developed biomimetic NPs containing cationic bovine serum albumin conjugated with siRNA and exosome-coated NPs to improve drug delivery to the lung pre-metastatic niche. A pre-metastatic niche is a tumor environment affecting the secondary organ because of metastasis from the primary organ. The exosomal NPs given by an IV increased the gene-silencing effects, inhibiting the growth of malignant breast cancer cells. The result of this study indicates that exosomes containing self-assembled NPs can be a promising strategy for suppressing postoperative breast cancer metastasis.

### 5.2. Inorganic NPs 

An NP that uses inorganic compounds such as gold, carbon, and iron as a carrier for drug delivery is called an inorganic NP. Inorganic NPs enhance the treatment efficacy of techniques such as photodynamic therapy and hyperthermia. These are the non-invasive techniques used for the treatment of cancer. Figure 2 depicts some subtypes of inorganic NPs.

In photodynamic treatment (PDT), light, a photosensitizer (PS), and molecular oxygen are utilized. When the photosensitizer is exposed to light of the proper wavelength, it becomes excited. It destroys malignant cells by releasing reactive oxygen species (ROS) [116]. Despite PDT’s many benefits, such as non-invasiveness, simple operation, and fewer side effects, the therapeutic efficacy of anti-cancer drugs is diminished due to issues with the systemic dispersion of photosensitizers and shallow light penetration [117]. Inorganic NPs can help overcome these shortcomings because of the large surface-to-volume ratio offered, which will increase the loading capacity of the PSs and improve the sustained delivery of the drug for the treatment of breast cancer.

Hyperthermia is apoptotic cell death triggered by heat. Surgical oncologists commonly employ this technique to treat malignancies while minimizing collateral harm to healthy tissue [118]. Hyperthermia can be generated by light irradiation, called photothermal therapy, and by exposing magnetic NPs to alternating magnetic fields [119]. The laser and magnetic fields are the sources of heat generation, which will induce tumor cell death. The common problems with hypothermia are the specific delivery and short circulation time of anti-cancer drugs. Inorganic NPs can overcome these shortcomings to deliver anti-cancer agents to treat breast cancer [120].

This article discusses seven distinct forms of inorganic NPs that can be applied to treat breast cancer.

#### 5.2.1. Gold NPs

Gold NPs, including nanospheres, nanorods, nanoshells, and nanocages of gold, have been developed and tested as drug delivery systems [121,122,123]. Gold NPs are practical and ideal inorganic formations for gene therapy, drug delivery, and imaging applications. They are biocompatible, non-toxic, and non-immunogenic [124]. The surface plasma resonance properties offered by gold NPs make them potential candidates as photothermal therapeutic agents for breast cancer treatment. Surface plasmon resonance occurs when photons of visible light with a specific angle of incidence excite electrons in the metal surface layer, which subsequently move parallel to the metal surface. When gold NPs absorb light, they generate heat from electron–photon and photon–photon interactions and improve the PDT treatment by causing the induction of hyperthermia in the tumor cells [125]. Since gold NPs’ maximal absorbance wavelength is in the visible range, infrared light passes through normal cells with little absorption [126], causing hyperthermia in tumor cells with less harm to healthy cells [127]. Gold NPs also enhance singlet oxygen ROS generation [128], which helps in increasing the efficiency of PDT treatment for breast cancer [129].

In addition to hyperthermia, gold NPs are even utilized as potential drug and gene delivery vehicles. In contrast to organic NPs, gold NPs can be better altered to provide site-specific drug release [130]. Gold NPs with a hydrophobic inner shell and a hydrophilic outer shell were developed and tested for IV drug delivery [131]. In this work, an acid-cleavable hydrazine linkage was used to attach the anti-cancer drug Dox to the inner shell. Folate was linked to the outer shell to act as a recognition molecule. With such a flexible design, drugs were prudentially delivered to 4T1 cells to achieve a targeted and acid-triggered drug release.

Naser Mohammed et al. [132] used gold NPs to make multi-walled carbon nanotubes, which could improve cancer treatment efficacy according to plasmonic photothermal therapy. This cancer-ablation technique combines radiative and non-radiative phenomena to rapidly convert photon energy into heat. In this study, injectable NPs were created to potentiate plasmon resonance and were tested for their ability to treat breast cancer cell lines (MCF-7). These gold and carbon nanotube conjugates showed 61.7–85.3% cytotoxicity in MCF-7 cell lines at a concentration of 25 µg/mL given via the IV route, with an irradiation period of 120 s, by using an NIR laser under a wavelength of 1064 nm. The results suggest that PDT is both concentration- and time-dependent and can be beneficial in breast cancer.

#### 5.2.2. Magnetic NPs

Magnetic NPs are used in biological applications because of their high field irreversibility, small size, and surface activity [133,134]. In vitro, magnetic NPs are used for magneto relaxometry, diagnostic separation, and selection. In therapeutic research, they have been utilized to create hyperthermia, increase active drug targeting, and help diagnostic applications such as nuclear magnetic resonance imaging [135]. Magnetic NPs with superparamagnetic characteristics are promising carriers for targeted delivery. Functionalized magnetic NPs can concentrate and store therapeutic molecules with an external magnetic field, allowing for drug administration to specific locations. The use of magnetic resonance imaging is suitable for monitoring and determining dose distribution [136]. Since the late 1970s, researchers have studied magnetically guided drug targeting.

Zanganeh et al. [137] demonstrated cancer cell death in the early mammary tumors by modulating iron levels with ferumoxytol (anemia medication). Ferumoxytol-treated macrophages elevated levels of mRNA associated with proinflammatory Th1-type responses. Frumoxytol, at a 10 mg Fe-kg^−1^ IV dose, significantly decreased aggressive adenocarcinomas in a mouse model, as shown by hematoxylin and eosin staining, bioluminescence imaging, and Prussian blue staining. The authors explained that ferumoxytol induces an antitumor M1 phenotypic response in immune cells, which is validated by a rise in proinflammatory M1 macrophages. In addition, the ferumoxytol therapy stimulated the creation of ROS, which kill cancer cells. 

#### 5.2.3. Carbon-Based NPs

In PDT, fullerene, carbon nanotubes, and graphene are the commonly utilized photosensitizers nanocarriers [138,139,140]. Attached covalently or non-covalently to functionalized carbon-based nanomaterials, PSs increase PDT solubility and biocompatibility [141]. Fullerenes are carbon-based nanomaterials that create ROS when irradiated at the correct wavelength [128]. Single- or multi-walled carbon nanotubes are other PDT nanocarriers. Carbon nanotubes are good PS nanocarriers in PDT due to their quick elimination, minimal cytotoxicity, simplicity of functionalization, and consistent endocytosis [128]. Due to their large surface areas, graphene NPs have significant therapeutic loading capacities for PS absorption in tumor cells [142].

Shi et al. [143] produced fullerene (C60)-iron oxide NPs and derivatized them with PEG, chlorine e6, and folic acid. These multifunctional injectable NPs were tested for PDT, radiofrequency thermal therapy, and magnetic targeting in in vitro MCF-7 breast cancer cells and in vivo breast cancer mouse models. In vitro PDT experiments at 16.0 g/mL of these NPs and 532 nm laser irradiation showed 31.3% viability, and radiofrequency thermal therapy at the same concentration with 13.56 MHz radiofrequency showed 36.9% viability. In vitro studies showed 18.8% viability in radiofrequency thermal followed by PDT. Individual PDT and PTT treatments produced 62.0% and 37.0% apoptosis in S180 breast cancer tumor-bearing mice, respectively, whereas combining them increased apoptosis to 96%.

Few studies have used carbon-based NPs for active PDT for breast cancer. The synergistic effects of carbon-based NPs and PSs can increase the effectiveness of PDT for breast cancer [138]. Fullerene cages, such as C60 [144] and carbon nanotubes [145], can serve as PSs in PDT applications and create ROS from photons by themselves [146]. Fullerene derivatives are competitive PSs for PDT or for preclinical treatment, as no extra PS is required to generate ROS, so they should be studied further for the treatment of breast cancer [147]. Despite their advantages, carbon nanotubes can produce asbestos-like irritation, which is hazardous. So their short-term and long-term toxicity must be investigated and understood [148].

#### 5.2.4. Quantum Dots

Quantum dots (QDs) are a class of fluorescent nanomaterials with interesting chemical and physical characteristics compared to organic dyes [149]. Due to their high quantum yields, effortless surface modification, and modifiable optical properties, they have been employed as multifunctional nanocarriers for PDT [150]. They are even excellent donors in applications involving fluorescence resonance energy transfer [151].

Using a spectrophotometric assay, Monroe et al. [152] evaluated the cellular uptake, cytotoxicity, and ROS production of graphene quantum dots (GQDs) coupled with methylene blue PS against MCF-7 breast cancer cells grown in vitro. This study found that MB improved cytotoxicity and ROS production compared to a 1:1 ratio of GQD to MB.

Ghanbari et al. [153] prepared glucosamine-conjugated GQDs for the breast-cancer-specific delivery of curcumin. They generated injectable graphene quantum dots using a green and simple oxidizing approach. With (N-ethyl-N′-(3-(dimethylamino)propyl) carbodiimide/N-hydrosuccinimide activators, glucosamine-GQDs were covalently functionalized and loaded with curcumin to create curcumin/glucosamine-GQDs. The nanocarrier containing curcumin exhibited a pH-dependent and sustained release profile, with a release of 37.0% at pH 5.5 and 17.0% at pH 7.4 after 150 h. The targeted injectable nanocarriers displayed excellent fluorescence at 488 nm in in vitro cellular uptake and a more pronounced cytotoxicity impact than non-targeted nanocarriers. The results indicated that multifunctional nano-assemblies have potential for breast cancer cell-targeted delivery.

#### 5.2.5. Silica NPs

Silica NPs can store and slowly release drugs [154]. Mesoporous silica NPs are made by polymerizing silica. They have a tunable pore size (which allows for high therapeutic drug loading capacity) [155], a large surface area to volume ratio, and an involuntary drug release and are easy to modify with different functional groups or ligands, allowing for actively targeted drug delivery [154]. PSs can be covalently bonded or entrapped onto silica NPs for PDT applications [128].

Mesoporous silica can enclose nanostructures for medication delivery. For near infrared (NIR)-responsive nanoscale drug delivery systems, gold nanorods are coated with mesoporous silica. Zhang et al. [156] described a novel therapeutic IV injectable platform based on mesoporous silica-coated gold nanorods loaded with Dox. The NIR light effectively causes Dox to be released from the loaded nanorods. The combination of Dox and the photothermal effect displayed near-infrared light and showed a better performance during the in vitro analysis against the tumor cells.

Similarly, Wang et al. [157] developed mesoporous silica NPs for IV cancer therapy using siRNA/ microRNA (miRNA) combinations. siRNA and miRNA can mute a specific group of oncogenes and target several disease pathways. Multifunctional tumor penetrating mesoporous silica NPs were prepared using PSs indocyanine green encapsulation and surface conjugation with an iRGD peptide for breast cancer treatment. In vitro 3D tumor spheroids and in vivo orthotopic MDA-MB-231 breast tumors demonstrated enhanced NP uptake in cells. Following short light irradiation (808 nm) of the primary tumor, systemic IV treatment with mesoporous silica NPs significantly suppressed primary tumor growth and reduced metastasis. Therefore, mesoporous silica NPs co-loaded with siRNA/miRNA are a potential candidate for treating breast cancer with metastasis.

#### 5.2.6. Ceramic NPs

Ceramic NPs are inorganic solids made of oxides, carbides, carbonates, and phosphates [154]. Porous ceramic NPs that can modulate drug release include silica, titanium dioxide, alumina, zirconia, calcium carbonate, and hydroxyapatite [154]. Ceramic NPs are stable, chemically inert, heat resistant, and easy to conjugate to hydrophilic or hydrophobic drugs [158]. They are ideal for drug delivery, imaging, dye photodegradation, and photocatalysis applications [154,159].

Platinum-loaded, selenium-doped hydroxyapatite NPs were created by Barbanente et al. to limit the growth of prostate and breast cancer cells without impacting the proliferation of bone marrow stem cells [160]. After seven days, the release kinetics of selenium and platinum from these NPs revealed cumulative release of 10 wt% and 66 wt%, respectively. At a platinum/selenium ratio of 8, the number of cells of MDA-MB-231 and PC3 were reduced by a factor greater than 10, with minimal effects on co-cultured human bone marrow stem cells. Due to their excellent anti-cancer selectivity, these NPs have the potential to be used as targeted chemotherapeutic agents against breast malignancies.

#### 5.2.7. Upconversion NPs

In PDT applications, upconversion NPs typically employ the light upconversion process for deep tissue penetration [161,162]. Upconversion is the conversion of near-infrared light to a visible light wavelength that has a shorter wavelength [161]. So up-conversion NPs (UCNPs) can absorb two or more low-energy photons and show an anti-stokes shift of the fluorescence emission in UV–Vis wavelengths (300–700 nm) when excited by NIR light (750–1400 nm) [163]. Due to their better reduced fluorescence background and lower phototoxicity, UCNPs have found use in biomedical applications [164]. UCNPs can excite PS electrons efficiently to produce efficient levels of singlet oxygen for PDT of deep-seated cancers under the fluorescence they emit at longer wavelengths [128,165].

UCNPs have recently been used as drug delivery carriers and PDT agents in developing UCNP-based therapy systems. UCNPs were designed as drug delivery vehicles using Dox as a model anti-cancer drug using various loading and releasing strategies [166,167,168,169]. For example, UCPs made with tocopherol polyethylene glycol 1000 succinate were used to deliver Dox [168]. Since tocopherol polyethylene glycol 1000 succinate can inhibit p-glycoprotein expression and ease intracellular drug administration, this nano system effectively killed Dox-resistant MCF-7 cells.

Zeng et al. [170] also showed that NIR PDT works with FA-functionalized, photosensitizer-loaded Fe_3_O_4_ Ytterbium- and Erbium-Doped Sodium Yttrium Fluoride (FA-NPs-PS) nanocomposites, both in vitro and in vivo. The effects of PDT on the viability of MCF-7 and Hela cells were initially examined in vitro. MCF-7 and HeLa cell viability was reduced to 18.4% and 30.7%, respectively, after 10 min of NIR irradiation (980 nm), while control viability was 93.9% and 91.3%, respectively. Furthermore, nude mice bearing MCF-7 tumors were used to test the in vivo therapeutic efficacy of FA-NPs-PS. After 15 days, the tumor volume in FA-NPs-PS treated with NIR irradiation decreased, but the control groups’ tumors grew dramatically. These results show that FA-NPs-PS nanocomposites and NIR-activated PDT could be used to treat MCF-7 tumors.

### 5.3. Other NPs

With the development of nanotechnology, the concerns associated with conventional therapies, such as non-specific and non-selective tissue damage, have been mitigated [171,172,173,174]. Nanocarriers can exhibit the targeted action of drugs due to their improved permeation and retention effects. Similarly, the microenvironment of tumors exhibits several physical and chemical characteristics, such as a redox state (increase in level of ROS), overexpressed proteins and enzymes, and hypoxia [171,175]. These features can regulate drug-loaded nanocarrier release.

#### 5.3.1. Reactive Oxygen Species (ROS)- Responsive NPs

ROS are a by-product of the mitochondrial process that is produced by the single electron reduction in oxygen in the body. The release of ROS occurs as a process of electron leakage before passing to the terminal oxidase respiratory chain, which consumes around 2% of the oxygen in the body [176]. ROS are produced and released internally during cellular respiration, metabolism, and enzyme activity. Oxygen molecules in a high-oxygen environment and the released mitochondrial respiratory chain reduce large electrons during state transition from III to IV [177]. Superoxide dismutase, catalase, glutathione, and thioredoxin neutralize ROS in normal physiological conditions. Nevertheless, when redox equilibrium is disrupted or eased, variations in ROS levels will result in physiological and pathological alterations [178,179]. ROS also regulate intracellular signal transduction pathways, constraining several biological channels to preserve cellular homeostasis. Any deviation from this sensitive equilibrium is followed by a cascade of events that can have beneficial or adverse effects on the cell.

Recent advancements in ROS biology demonstrated that their impacts are gradient-dependent, resulting in “good” or “bad” ROS [180]. Through many signaling pathways, abnormal ROS production and insufficient neutralization can help tumors grow and spread [181]. An efficient, integrated therapeutic system can be designed to deliver chemotherapeutic drugs on demand and reduce ROS production.

#### 5.3.2. Enzyme-Responsive NPs

A tumor microenvironment’s metabolism is abnormal because of the overexpression of the matrix metalloproteinase enzyme. Enzymes such as calcium- and zinc-dependent endopeptidases play a huge role in cancer cell extracellular matrix remodeling [182]. Additionally, enzymes such as lysyl oxidase and hyaluronidase are also overexpressed in tumors, which can be exploited for achieving the targeted delivery of NPs [183]. 

Kashyap et al. [184] co-polymerized a hydrophobic acrylate monomer with polyethylene glycol. This copolymer core-shell NPs system was employed to distribute Dox-responsive nanoscaffolds for cancer treatment. In the presence of esterase (pH = 7.4, 37 °C), the amphiphilic copolymer decomposed slowly and regularly after 12 h, releasing > 95% of the drugs and achieving controlled release in cells.

#### 5.3.3. Hypoxia-Activated NPs

The oxygen partial pressure is typically below 60 mmHg and falls as the tumor grows. Tumor proliferation increases the diffusion distance between blood vessels and tumor cells, resulting in decreased oxygen delivery and hypoxia. The hypoxic environment promotes tumor development, multiplication, and metastasis and even diminishes the efficacy of chemotherapeutic drugs [185].

Mpekris et al. [186] observed that the lung metastasis of breast cancer constricts blood vessels, causing hypoxia. In this study, three different drugs, tranilast, atezolizumab, and the nano-drug Doxil, were given. Tranilast treatment was given by decompressing lung metastatic blood vessels in mice to restore perfusion and reduce hypoxia. It was then easier for atezolizumab and the nano-drug Doxil to show the cytotoxic effect on metastases and start an immune response against cancer.

## 6. Conclusions

This review discusses the conventional breast cancer therapies, their shortcomings, and how injectable nanoparticulate systems can help address these shortcomings and improve the therapeutic effectiveness of the drugs used in breast cancer treatment. Recent advances in organic, inorganic, and NPs-based delivery systems’ targeting, based on their surface modification, and the tumor microenvironment such as hypoxia, redox state (increase in the level of ROS), and enzyme production are also discussed. NPs have a promising role in increasing the therapeutic efficacy toward breast cancer treatment both in vitro and in vivo. NPs work via multiple mechanisms and can be used for both active and passive targeting. In addition to drug targeting, nano-formulations can improve some of the physiochemical challenges of anticancer agents and minimize their systemic toxicity. Although the oral route is the most preferred route of drug administration; in the case of breast cancer treatment, the injectable route can be preferrable because of certain advantages. IV administration provides an instantaneous response, complete bioavailability, and avoidance of the first-pass metabolism of drugs. Surface modification with stealth properties and the ability to target tumor cells by active targeting moieties on the surface enhances therapeutic effectiveness and minimizes toxicity. The outstanding performance of NPs can be attributed to their small size and a high degree of uniformity. However, the large-scale synthesis of NPs while preserving their uniformity remains challenging. Currently, developing NPs to deliver drugs at the metastasis site where leaky vasculature is absent needs an active targeting strategy, which can be a challenging and daunting task for a formulation scientist [187,188].

Several nanomedicine products are currently available in the market for the treatment of breast cancer. However, preclinical-to-clinical translation remains a great challenge. This is due to the limited knowledge of the mechanisms of interaction of nanomaterials with various tissues and organs in the body. Additionally, oversimplification of the EPR effect, a poor correlation between the in vivo animal data and the clinical translation in humans, differences in the microenvironment of the tumor between animals and humans, and their heterogeneity are responsible for the gap in the translation of nanomedicine to clinical practice [189]. However, by developing a patient-focused, disease-driven design for nanomedicine, with robust in vitro and ex vivo models sufficiently capable of extrapolating the animal data to humans, it is possible to overcome these translational challenges from the bench to the bed side. Many more innovative approaches are expected to be developed in the near future to address these translational challenges.

## Figures and Tables

**Figure 1 pharmaceutics-14-02783-f001:**
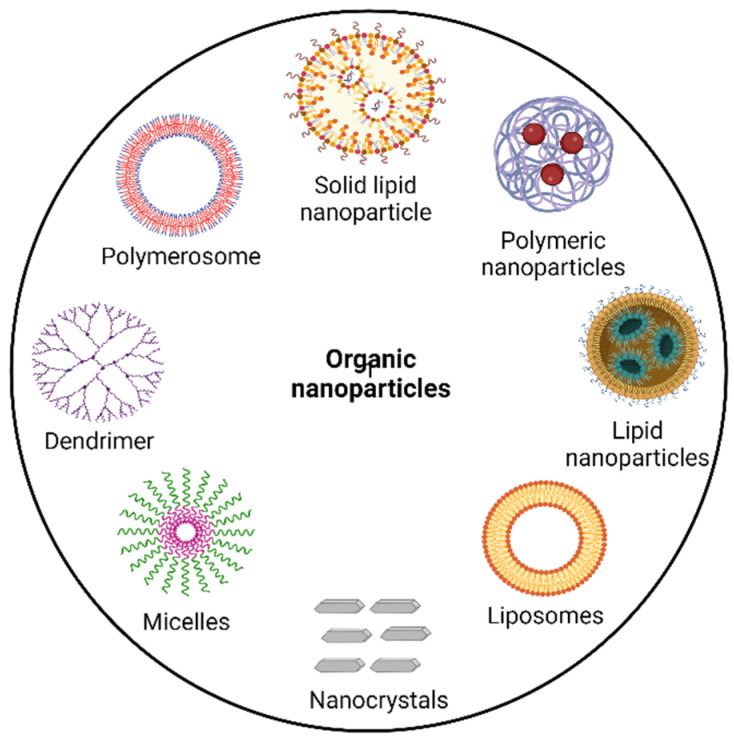
Important types of organic nanoparticles.

**Figure 2 pharmaceutics-14-02783-f002:**
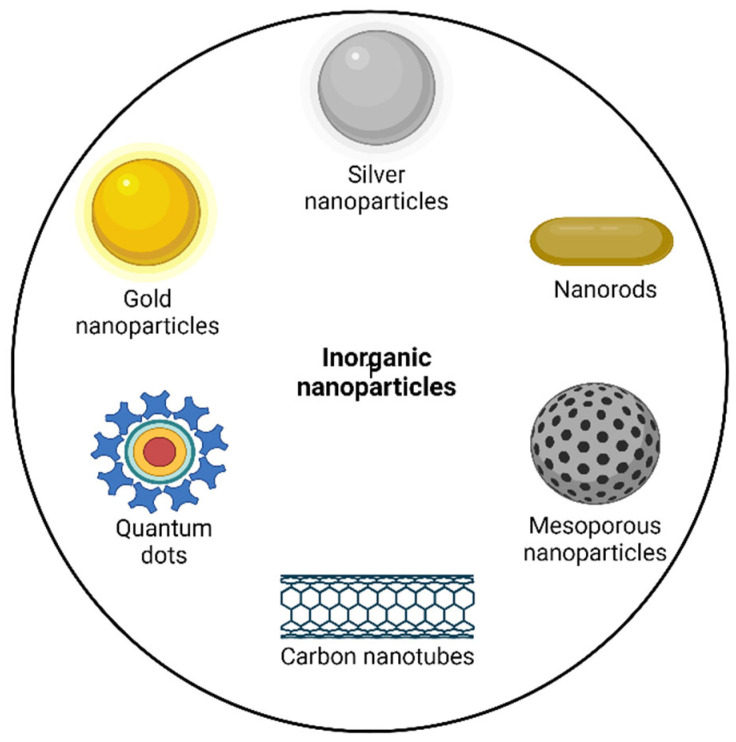
Important types of inorganic nanoparticles.

**Table 1 pharmaceutics-14-02783-t001:** Subtypes of breast cancer [18,19,20,21].

Types of Breast Cancer	Receptor	Subtypes	Commonly Used Medications
Hormone positive	ER^+^ or PR^+^	Luminal A and B	Tamoxifen, Exemestane, Anastrozole, Letrozole
HER2-positive	HER2^+^	-	Trastuzumab, Lapatinib, Neratinib, Pertuzumab
Triple-negative	ER^-^, PR^-^, and HER2^-^	Basal-like 1, basal-like 2, immunomodulatory, mesenchymal, mesenchymal stem cell like, and luminal androgen receptor	Paclitaxel, Doxorubicin, 5-fluorouracil, Eribulin, Docetaxel

**Table 2 pharmaceutics-14-02783-t002:** Marketed chemotherapeutic agents with their doses for breast cancer treatment.

Chemotherapeutic Drugs	Dose	Reference
**Paclitaxel**	**Node Positive**: 175 mg/m^2^ IV over 3 h q3 weeks 4 times (with doxorubicin (Dox) regimen) **Metastatic Disease (failure of initial chemotherapy or relapse within 6 months following adjuvant chemotherapy)**: 175 mg/m^2^ IV over 3 h q3 weeks	[51]
**Dox**	60–75 mg/m^2^ IV q21 Days	[52]
**Epirubicin**	Day 1: Epirubicin 100 mg/m^2^ IV, 5-fluorouracil 500 mg/m^2^ IV, and cyclophosphamide 500 mg/m^2^ IV followed by repetition up to q21 days × 6 cycles	[53]
**Docetaxel**	**Locally Advanced or Metastatic** For locally advanced or metastatic breast cancer after failure of prior chemotherapy Monotherapy: 60–100 mg/m^2^ IV over 1 h q3 Weeks **Dosage Modifications (adjuvant treatment)** Initial: 75 mg/m^2^ Reduced to 60 mg/m^2^ in patients with febrile neutropenia treated with G-CSF, severe or cumulative cutaneous, or neurosensory reactions	[54]
**Fluorouracil**	500 or 600 mg/m^2^ IV on days 1 and 8 q28 days for 6 cycles as a component of a cyclophosphamide-based multidrug regimen	[55]
**Cyclophosphamide**	600 mg/m^2^ IV with other antineoplastics	[56]
**Albumin-bound paclitaxel**	260 mg/m^2^ IV infused over 30 min q3 weeks	[57]
**Eribulin**	1.4 mg/m^2^ IV infused over 2–5 min on days 1 and 8 of 21-day cycle	[58]

**Table 3 pharmaceutics-14-02783-t003:** Approved breast cancer drug therapies via parenteral route based on nanotechnology.

Year	Product	Nanoparticle (NP) Material	Drug	Indication	Company	Reference
2007 (S. Korea)	Genexol-PM	PEG-PLA polymeric micelle	Paclitaxel	Breast cancer, lung cancer, ovarian cancer	Samyang/Biopharm	[70]
2005	Abraxane	NP-bound albumin	Paclitaxel	Breast cancer, pancreatic cancer, non-small cell lung cancer	Abraxis/Celgene	[70]
2000 (EU)	Myocet	Liposome	Dox	Breast cancer	Cephalon	[70]
1998	Lipo-Dox	Liposome	Dox	Kaposi sarcoma, breast cancer, ovarian cancer	Taiwan Liposome	[70]
1995 1999 2003 2007 (Europe, Canada)	Doxil	Liposome	Dox	Kaposi sarcoma, ovarian cancer, breast cancer, multiple myeloma	Johnson and Johnson	[70]

**Table 4 pharmaceutics-14-02783-t004:** Current clinical studies investigating nanomedicine to treat breast cancer.

NCT Number	Titles Conditions		Interventions	Phases	References
NCT01583426	Nanoparticle-based paclitaxel vs, solvent-based paclitaxel as part of neoadjuvant chemotherapy for early breast cancer	Tubular breast cancer stage II and III HER2+ breast cancer Invasive ductal breast cancer	Drug: Nab-Paclitaxel Drug: Paclitaxel	Phase 3	[71]
NCT03749850	Image-guided targeted doxorubicin delivery with hyperthermia to optimize loco-regional control in breast cancer	Metastatic breast cancer Invasive ductal breast cancer Adenocarcinoma breast	Drug: Liposomal Doxorubicin Procedure: Magnetic-resonance-guided high-intensity-focused ultrasound Drug: Cyclophosphamide	Phase 1	[72]
NCT00046527	Study of ABI-007 and Taxol in patients with metastatic breast cancer	Breast neoplasms Metastases, neoplasm	Drug: ABI-007	Phase 3	[73]
NCT01730833	Pertuzumab, Trastuzumab, and Paclitaxel albumin-stabilized NP formulation in treating patients with HER2+ advanced breast cancer	HER2+ breast cancer Breast adenocarcinoma	Biological: Pertuzumab Biological: Trastuzumab Drug: Paclitaxel albumin-stabilized NP formulation Other: Laboratory biomarker analysis	Phase 2	[74]
NCT00915369	A clinical trial to study the effects of an NP-based paclitaxel drug, which does not contain the solvent cremophor, in advanced breast cancer	Advanced breast cancer	Drug: Nanoxel (Paclitaxel NP formulation)	Phase 1	[75]

## Data Availability

Not applicable.

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
