# Peer review of "Injectable Nano Drug Delivery Systems for the Treatment of Breast Cancer"

_pharmaceutics, 2022, doi:10.3390/pharmaceutics14122783_

Round 1
Reviewer 1 Report
The article is well written but based on the review article title - Injectable drug delivery systems for breast cancer treatment, I would like to suggest authors to focus majorly on injectable formulations rather than all formulations studied for the treatment of cancer.
I would recommend its publication after addressing the following comments.
Also, please mention the route of administration while discussing about the formulations and try to summarize the results obtained with cell line details/type of cancer.
My comments are as follows:
1. HER2-positive should be HER2+ (line 80).
2. Include reference in line 96/97.
3. In table 2/3/4, please add references (as a separate column against each statement).
4. In table 3/4 - Kindly focus on breast cancer rather than all types of cancer.
5. Mention route of administration in each description/discussion.
6. Mention the treatment details like cell lines used/category of breast cancer and its toxicity results/profile.
7. Mention the source of light and its wavelength under PTT/PDT discussion.
8. Authors should give emphasis majorly on the injectable formulations rather than general nano-formulations for breast cancer treatment.
9. Please include one separate section and discuss about the comparison/advantages of injectable formulations over other routes of administration for the treatment of breast cancer.
Author Response
Reviewer #1
1: The article is well written but based on the review article title - Injectable drug delivery systems for breast cancer treatment, I would like to suggest authors to focus majorly on injectable formulations rather than all formulations studied for the treatment of cancer.
I would recommend its publication after addressing the following comments.
Also, please mention the route of administration while discussing about the formulations and try to summarize the results obtained with cell line details/type of cancer.
Response: We thank the reviewer for his/her complements. We appreciate the time and effort you have dedicated to critically reviewing and providing insightful comments and suggesting valuable improvements to this manuscript. We have modified and incorporated all the suggestions. Please see below our detailed response to each of thecomments. All page numbers refer to the manuscript file with track changes.
2: HER2-positive should be HER2+ (line 80).
Response: We have changed HER2-positive to HER2+ in line 80.
3: Include reference in line 96/97.
Response: We have added reference no [30] for early menarche section in line 96/97 as per the suggestion.
4: In table 2/3/4, please add references (as a separate column against each statement)
Response: We have added a separate column for references in table 2, 3, and 4.
5: In table 3/4 - Kindly focus on breast cancer rather than all types of cancer.
Response: We have deleted the examples related to other cancers from table 3 and 4 and have only included examples related to breast cancer.
6: Mention route of administration in each description/discussion and try to summarize the results obtained with cell line details/type of cancer.
Response: We have included routes of administration and cell line details at relevant places.
7: Mention the source of light and its wavelength under PTT/PDT discussion.
Response: We have included source of light and its wavelength at relevant places.
8: Please include one separate section and discuss about the comparison/advantages of injectable formulations over other routes of administration for the treatment of breast cancer.
Response: We have included one paragraph discussing about the comparison/advantages of injectable formulations over other routes of administration for the treatment of breast cancer from line 233-241 at page no. 9. This paragraph is followed by discussion of the advantages of injectable nanoparticles.
9: Authors should give emphasis majorly on the injectable formulations rather than general nano-formulations for breast cancer treatment.
Response: We have minimized discussion related to general nano formulations and have focused more on injectable formulations for the treatment of breast cancer in the revised manuscript.

Reviewer 2 Report
In the present manuscript entitled “Injectable Nano Drug Delivery Systems for the Treatment of Breast Cancer” the authors have nicely compiled the information for novice researchers to have a basic understanding of the disease as well as the nanotechnology-based treatment options currently available or under investigation. Exosomes isolated from various sources especially cow’s milk are under intensive investigation and have shown promising results when delivered orally. The authors are suggested to include one paragraph on exosomes as well to make the readers aware of this emerging technology. In a previous report on milk exosomes, authors tested the milk exosomes for oral delivery of paclitaxel and reported excellent efficacy in the breast cancer tumor xenograft model (https://www.sciencedirect.com/science/article/pii/S1549963417300436). There are several other reports as well in which exosomes isolated from cow’s milk or other sources have shown their potential as a delivery vehicle for a variety of natural agents, genetic materials, and chemotherapeutics either alone or in combination. Authors can consider making a separate table compiling different nanoparticles including exosomes which could not be covered during the discussion. The authors are advised to include only breast cancer-related clinical trials in Table 4 or if they don’t find a sufficient number of ongoing clinical trials to justify a separate table then it can be merged with Table 3. The authors' expert commentary is highly welcomed about the pros and cons of the different nano drug delivery systems.
Author Response
1: In the present manuscript entitled “Injectable Nano Drug Delivery Systems for the Treatment of Breast Cancer” the authors have nicely compiled the information for novice researchers to have a basic understanding of the disease as well as the nanotechnology-based treatment options currently available or under investigation.
Response: Thank you for reviewing the manuscript and providing the valuable suggestions and positive feedback.
2: Exosomes isolated from various sources especially cow’s milk are under intensive investigation and have shown promising results when delivered orally. The authors are suggested to include one paragraph on exosomes as well to make the readers aware of this emerging technology. In a previous report on milk exosomes, authors tested the milk exosomes for oral delivery of paclitaxel and reported excellent efficacy in the breast cancer tumor xenograft model (https://www.sciencedirect.com/science/article/pii/S1549963417300436). There are several other reports as well in which exosomes isolated from cow’s milk or other sources have shown their potential as a delivery vehicle for a variety of natural agents, genetic materials, and chemotherapeutics either alone or in combination. Authors can consider making a separate table compiling different nanoparticles including exosomes which could not be covered during the discussion.
Response: Thank you for your suggestions on exosomes. We have developed one section on exosomes as injectable drug delivery systems for treatment of breast cancer from lines 395-420 on pages 14 and 15. We have included the reference as suggested but since this work was based on exosomes for oral delivery, we have included other references that studied exosomes for injectable delivery in the treatment of breast cancer.
Likewise, we have also included a discussion on drug nanocrystals on page 14 which was not covered during the previous version of the manuscript.
3: The authors are advised to include only breast cancer-related clinical trials in Table 4 or if they don’t find a sufficient number of ongoing clinical trials to justify a separate table then it can be merged with Table 3.
Response: As suggested, we have excluded the examples related to other cancers and updated tables 3 and 4, including only those relevant to breast cancer.

Round 2
Reviewer 2 Report
The revision is satisfactory and the revised version can be accepted for publication.